# Internal thread defect detection system based on multi-vision

**Xiaohan Dou**[☯], **Chengqi Xue**[☯], **Gengpei Zhang**[iD]*, **Zhihao Jiang**

Yangtze University, Jingzhou, Hubei, China

☯ These authors contributed equally to this work.
* judgebill@126.com

**Data Availability Statement:** All relevant data are within the manuscript and its Supporting Information files.

**Funding:** The author(s) received no specific funding for this work.

## Abstract

In the realm of industrial inspection, the precise assessment of internal thread quality is crucial for ensuring mechanical integrity and safety. However, challenges such as limited internal space, inadequate lighting, and complex geometry significantly hinder high-precision inspection. In this study, we propose an innovative automated internal thread detection scheme based on machine vision, aimed at addressing the time-consuming and inefficient issues of traditional manual inspection methods. Compared with other existing technologies, this research significantly improves the speed of internal thread image acquisition through the optimization of lighting and image capturing devices. To effectively tackle the challenge of image stitching for complex thread textures, an internal thread image stitching technique based on a cylindrical model is proposed, generating a full-view thread image. The use of the YOLOv8 model for precise defect localization in threads enhances the accuracy and efficiency of detection. This system provides an efficient and intuitive artificial intelligence solution for detecting surface defects on geometric bodies in confined spaces.

## 1. Introduction

Threaded fittings hold significant importance across various industrial sectors, including aerospace, automotive manufacturing, and beyond, serving as crucial components for connecting and securing diverse elements. The effectiveness of their seal directly influences the potential leakage of liquids or gases, posing a substantial risk for safety incidents and environmental pollution in multiple industries. Consequently, developing and implementing robust safety testing methods is vital to guarantee the integrity and operational efficiency of these threaded connections.

Internal threads, typically situated in narrow and deep spaces, are challenging to examine using traditional observational and measurement tools, complicating the detection of defects. The wide array of potential defects, such as surface and fatigue cracks, impurities, and pores, with their varied shapes and sizes, necessitates highly accurate and sensitive methods for precise identification and assessment. Furthermore, the rapid and efficient detection of these defects is a critical concern in industrial production lines where large volumes of threaded fasteners need swift and precise inspection. Conventional manual inspection methods, being time-consuming and prone to errors, fall short of meeting these efficiency demands.

**Competing interests:** The authors have declared that no competing interests exist.

Additionally, the lack of natural light in internal threads presents unique challenges in designing internal lighting solutions. Consequently, the development of an efficient, automated system for internal thread defect detection has emerged as an imperative requirement.

In recent years, research on external thread detection has advanced considerably. Detection methods can be broadly classified into two categories: contact and non-contact. Contact methods encompass techniques such as thread gauge detection (qualitative), measuring needle detection (including two-needle and three-needle methods), thread micrometer inspection (single inspection), sample inspection, three-coordinate measurement [1], and contact scanning. Although these methods tend to be more accurate, they require skilled inspectors using specialized tools for precise operation, leading to higher labor costs and lower efficiency. Additionally, they are susceptible to subjective bias and may cause wear on the workpiece during inspection. In contrast, non-contact methods utilize optical principles, including techniques like optical microscope inspection [2], laser scanning [3], and machine vision [4]. These methods simplify certain manual tasks and complex calculations through high-precision inspection equipment, enabling automated control and enhanced accuracy. Machine vision-based defect inspection, leveraging images captured by specific light sources and hardware, significantly improves the quality, efficiency, and reliability of defect detection. However, due to the influence of the diffraction limit, optical microscopes have inherent limitations in resolution, and the depth of field of optical microscopes is usually shallow. For thicker samples, it is impossible to keep the entire volume in a clear focal plane simultaneously. For sensitive materials, the use of strong lasers may cause surface damage or changes to the sample, thereby affecting the accuracy of detection results. Compared with the previous two defect detection methods, the defect detection method based on machine vision can quickly process and analyze images, achieving high-speed automatic detection, significantly improving the detection speed and efficiency of the production line, and is suitable for various types and complexities of defect detection tasks.

The inspection of internal threads presents a multifaceted technical challenge, primarily due to the confined spaces, inadequate lighting, and the intricate geometry of the threads themselves, as depicted in Fig 1. Addressing these challenges necessitates innovative solutions in both hardware and software. For hardware, the development of more compact and adaptable detection equipment is essential to navigate the constraints of limited space. Enhanced lighting solutions are also crucial to provide sufficient illumination in environments lacking natural light, thereby ensuring clear image acquisition. On the software front, sophisticated image processing techniques are vital for accurately discerning thread characteristics in images beset by low illumination and high noise levels.

For defect detection of internal threads, high-precision laser sensors [5] can be used. Through the integration of high-precision optical detection and ranging (LiDAR) technology and cameras [6], accurate detection of structural cracks has been achieved. This method can be applied to the detection of defects in internal threads to verify their quality. However, this method does not perform well on reflective surfaces or transparent materials, as the laser might be scattered or refracted, affecting the accuracy of measurements. Image-based systems, utilizing linear lasers and template matching techniques [7], can quickly detect thread defects. Yet, for parts with complex or non-standard thread shapes, it may be necessary to create multiple templates, increasing the system's complexity and computational demands. Utilizing novel arrayed differential flexible eddy current sensors [8], defects on the surface of iron threads can be detected, but the detection effectiveness for non-metallic materials is limited, and the performance of eddy current sensors may be influenced by the conductivity of the test materials. Through digital image processing techniques [9], OTPG can standardize and segment internal thread images, further assessing their quality. This method depends on the quality of images, and the accuracy of the processing algorithms decreases in conditions of insufficient light or

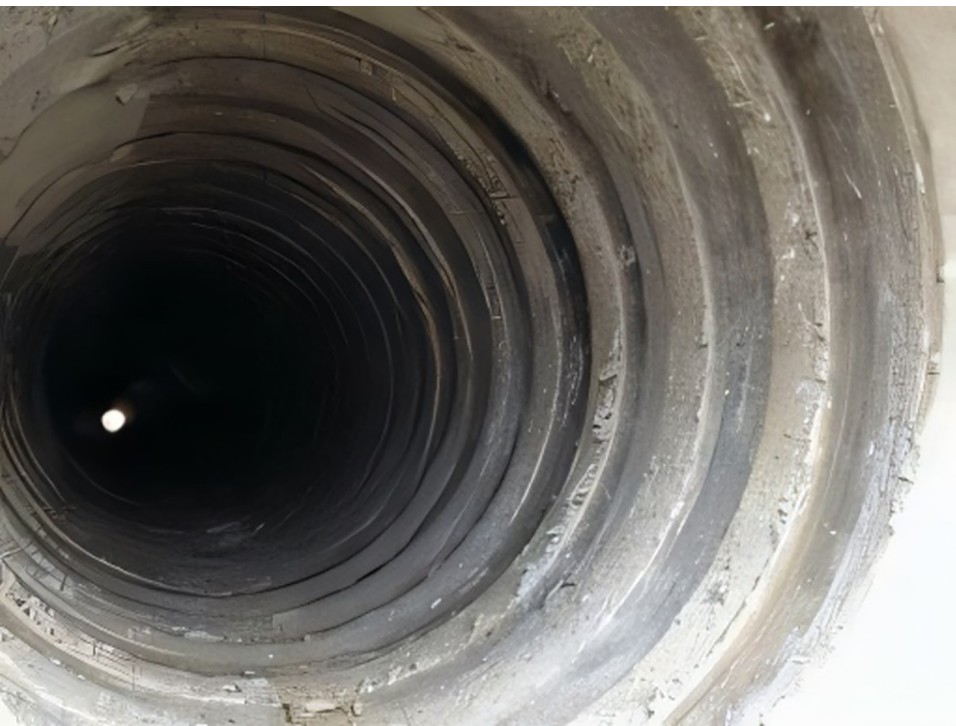

**Fig 1. Internal thread pipe.**

poor image quality. Moreover, complex image processing algorithms require higher computational resources and processing time, affecting the real-time nature of detection.

While the aforementioned methods enable the detection of internal thread defects, they are not without their limitations. The current state of research in machine vision, recognized for its efficacy in defect detection, is as follows: Machine vision has achieved high accuracy and precision in detecting weld defects in thin-walled metal cans [10]; it is effectively employed in the detection of traps in printed circuit boards [11], an approach that is independent of color, position, and direction, enabling rapid and accurate defect identification; similarly, machine vision is extensively utilized for detecting surface defects in iron and steel products [12]. Real-time defect detection of metal components [13] has significantly enhanced the accuracy and speed of identifying surface flaws and measuring dimensions in metal parts by employing YOLOv6 and an improved Canny–Devernay subpixel edge detection method. For internal thread defect detection, machine vision, compared to other methods, maintains detection speed while ensuring the accuracy of the detection results.

This paper delves into the technologies used for detecting both internal and external threads, presenting a well-conceived lighting design solution to address the illumination challenges inherent in internal thread detection. Furthermore, it employs advanced machine vision and image processing technologies to enable an automated and highly efficient approach for pinpointing defects in internal threads, thereby significantly enhancing the overall detection efficiency.

## 2. System theory

This paper's methodology is bifurcated into two distinct components: hardware and software, as depicted in Fig 2. At the core of the system lies the Raspberry Pi microcomputer, tasked with controlling the camera for image capture, maneuvering the stepper motor, and operating

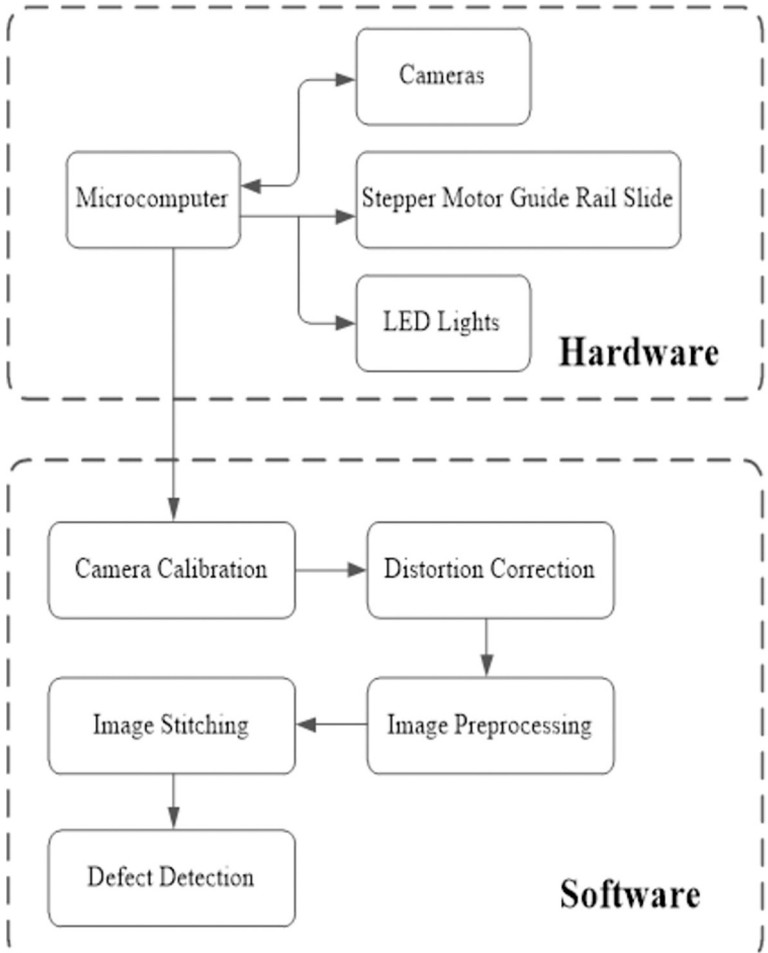

**Fig 2. Flow chart of internal thread defect detection.**

the lighting system. Post-image acquisition, the camera undergoes calibration, facilitating the correction of distorted images based on the acquired calibration parameters. Subsequently, these rectified images are processed and seamlessly stitched together, culminating in the detection of the final amalgamated image.

Given the limited lighting distance and the lack of natural light within internal threads, the lighting equipment for this study's application scenarios has been specifically designed. The vision acquisition system employs a fisheye camera to minimize the number of cameras required and to afford a single camera a wider field of view. The restricted diameter of the internal thread constrains the shooting angles of adjacent cameras, preventing them from covering a common area and thus posing challenges in image splicing. To address this, we propose a hardware architecture strategy that involves staggered and overlapping camera shots, effectively overcoming these splicing challenges.

### 2.1. Image acquisition

Owing to the unique structural features of internal threads, their non-contact inspection necessitates addressing the challenge of illumination and facilitating the placement of the vision system within the threaded fittings.

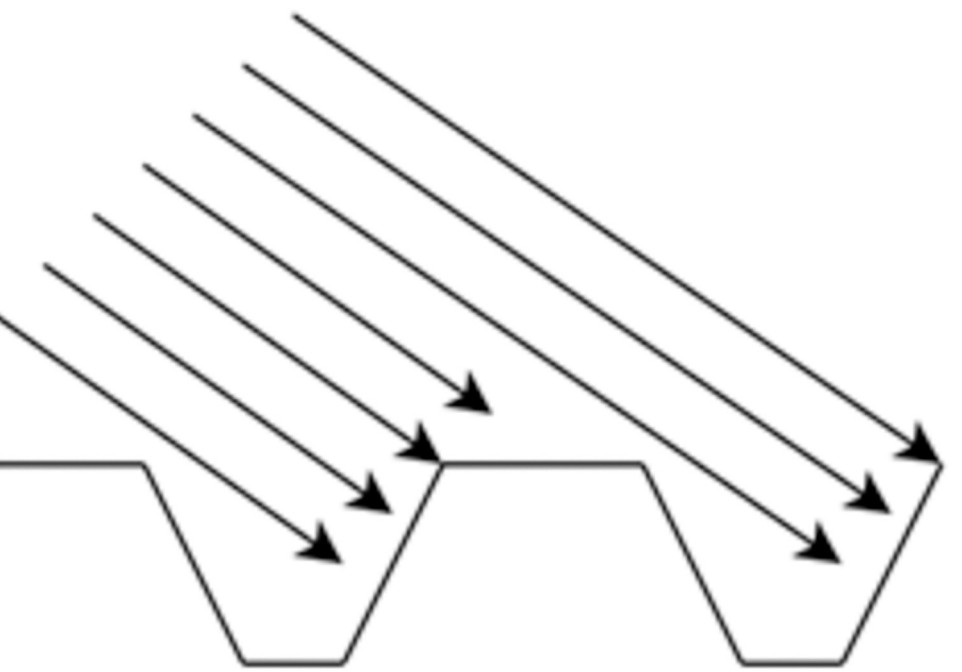

**Fig 3. Parallel light shines on the thread.**

In designing the illumination system, we focused on key aspects such as the selection of the light source, as well as considerations of field of view and depth. Our findings revealed that illumination from a point light source results in shadows within the wide-angle camera's field of view, due to the protruding threads partially obscuring the light, as demonstrated in Fig 3.

Consequently, our experimental findings and debugging efforts have led us to recommend a specific operational method for capturing images of internal threads, as illustrated in Fig 4.

This study opts for an LED strip light source, positioned on the edge side of the visual system, over a point light source LED. The LED strip light emits a softer light. When positioned at the correct incidence angle, it enables shadow-free visualization within the fisheye lens's field of view. Given its specific focal length range, the fisheye lens facilitates the acquisition of images across various sizes of threaded pipe fittings, laying the groundwork for subsequent image splicing. The fisheye lens also captures a broader range of information. Consequently, we employ six fisheye lenses to create a comprehensive visual system. These lenses are organized into two sets of three, arranged in a 120° configuration relative to each other, with the

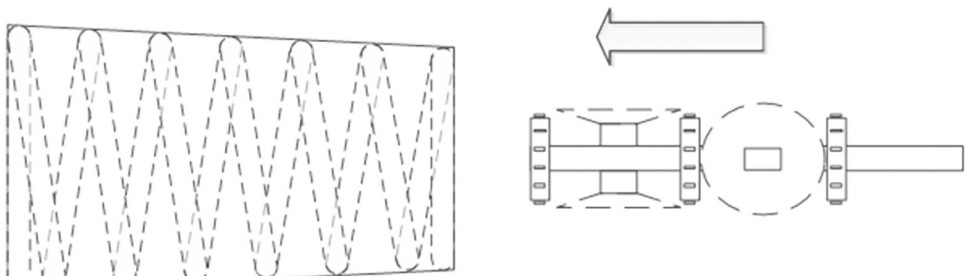

**Fig 4. Visual system model and how it works.**

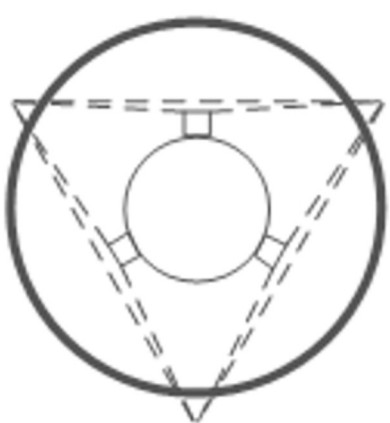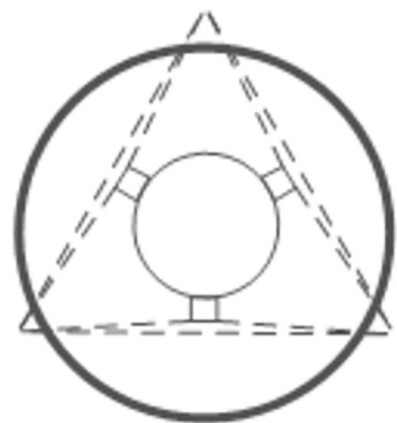

**Fig 5. Fisheye lens field of view model.**

two sets positioned in a front-to-back arrangement. The lenses within each set are offset by 60˚, ensuring complete coverage of the internal thread area by the system's field of view. To capture the entirety of the threaded pipe, the Raspberry Pi controls the stepper motor to adjust the vision system's position within the pipe, thus completing the image acquisition process.

In practical tests, we discovered that for some threaded pipe fixtures with small inner diameters, the camera's field of view did not cover the entire area inside the fixture, leaving segments of uncollected information, as shown in Fig 5. This lack of coverage indicates potential issues with thread splicing and defect detection in the missing areas.

To address image completeness, we improved the operational approach by employing stepper motors for finer control, as depicted in Fig 6. This method involves positioning the first camera group inside the threaded pipe fittings to cover the targeted internal thread area, as indicated in the figure. Once the first group completes its capture, the second group is positioned to capture the overlapping regions, facilitating seamless image splicing and defect detection.

## 2.2. Image distortion correction

Image correction involves selecting an appropriate vision system to define geometric aberrations, estimating parameters that describe image formation, and ultimately correcting image aberrations based on the camera model.

Utilizing an LCD panel for camera calibration [14] presents a notable advancement over conventional calibration techniques, achieving an impressively low average reprojection error of just 0.018. This method not only delivers high accuracy but also simplifies the calibration process by eliminating the need for manual intervention or additional equipment for movement adjustments. The implementation of the VDPC model [15] for wide-angle fisheye images introduces an innovative approach to correction. This model adaptively selects correction planes for various regions within the image, ensuring a uniform perspective correction across different areas of fisheye images and maintaining detail integrity with enhanced flexibility compared to the traditional Perspective Projection Conversion (PPC) method. Further advancements are achieved through the development of a new dataset, SLF [16], enabling the application of the Linear Perceptual Correction Network (LaRecNET) specifically designed to address fisheye image correction by exploiting the linear geometry inherent in fisheye images. The OmniVidar method [17], known for its precision in omnidirectional depth estimation, effectively tackles fisheye image distortion in a multitude of scenarios. To overcome the

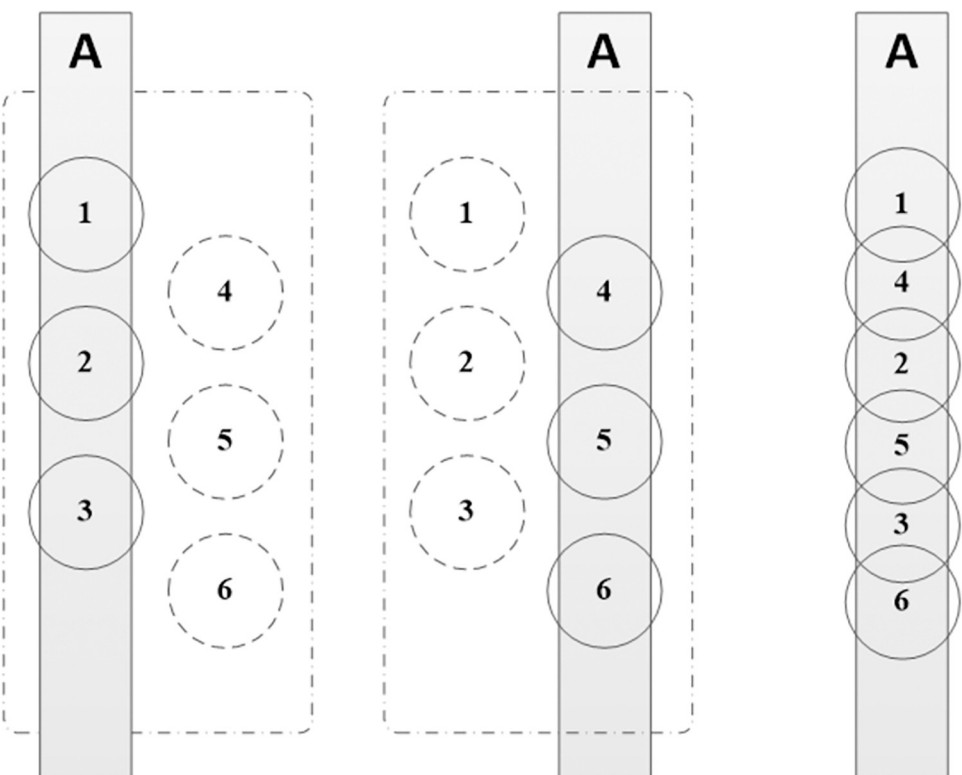

**Fig 6. Working mode diagram.**

limitations of existing models, a novel camera model, termed the three-sphere model, has been developed. This model revolutionizes the fisheye camera into a multi-vision system, significantly streamlining computational efforts. With this model, the necessity for a separate distortion correction step [18,19] is eliminated, allowing for direct depth estimation and image processing. The construction of the Fisheye DistanceNet framework [20] marks a further innovation, designed to predict the scale-aware distance of objects within fisheye camera images. Through extensive experimental evaluation across multiple datasets, this method has demonstrated superior accuracy in estimating object distances, showcasing enhanced performance over existing techniques.

This section outlines the process of camera calibration using a checkerboard pattern and a fisheye-specific correction approach, setting the groundwork for advanced image processing tasks.

## 2.3. Image stitching

The fundamental process of image stitching encompasses several stages: preprocessing the captured images, registering these images, fusing them, and ultimately producing the final stitched image. Challenges arise due to the imaging system's limitations and the noise introduced during image acquisition and transmission, which can obscure relevant thread features. To mitigate these effects, noise reduction is achieved through filtering techniques such as mean, median, Gaussian, and bilateral filtering, thus preserving feature integrity for accurate registration.Subsequent to noise reduction, image enhancement is applied to refine image quality, detail visualization, and overall appearance. This renders the images clearer and more conducive to analysis, with algorithms like the Retinex technique [21], Histogram Equalization [22], and Adaptive Histogram Equalization [23] being particularly effective.

Image registration leverages feature point extraction algorithms—namely Harris [24], FAST [25], SIFT [26], SURF, and ORB—with Harris demonstrating superior accuracy in identifying corners and contours, which is pivotal for measuring external thread parameters.However, direct stitching can introduce gaps or distortions at image junctures. Image fusion methods [27], including direct fill, average, Poisson fusion [28], weighted average [29], and multi-exposure fusion [30,31], are employed to address these issues. Each method has its advantages, from the simplicity and completeness of direct fill to the detailed, natural results of Poisson fusion, albeit with higher computational demands and complexity in adjusting for varied scenes. Weighted average and multi-exposure fusion techniques offer balanced solutions, improving image quality through careful weighting and detailed analysis, respectively, though they come with their own computational and procedural complexities.

This revision aims to streamline the explanation, emphasize the sequential steps in image stitching, and clarify the rationale behind the choice of techniques for noise reduction, image enhancement, and fusion, providing a cohesive overview of the process.

### 2.4. Image defect detection

This section delves into the algorithms for detecting defects within the stitched 2D images of internal threads. It explores various methodologies, including statistical [32], spectral [33], model-based, and pattern-based approaches [34]. Statistical methods can characterize the spatial relationship between pixels, providing useful information on regularity, roughness, self-similarity, and uniformity, yet they struggle to precisely identify and locate minor or atypical defects in internal threads. Spectral methods analyze pattern textures using different transformed domains, offering multi-resolution images to reduce computational costs, but they may fail to capture all necessary texture features in the high-frequency details of internal threads with complex geometric structures. Model-based methods characterize the linear dependency between different pixels in pattern texture images and capture local context information, but perform poorly when dealing with the irregular surfaces of internal threads. Pattern-based approaches, which view textures as combinations of textural primitives for feature extraction and defect detection, struggle to accommodate the complex three-dimensional structure of threads and the diversity of defects. For internal thread defect detection, the following methods are primarily considered: Fast r-cnn [35], based on prior work, uses deep convolutional networks to efficiently classify object proposals, significantly improving speed. Ssd [36], based on a single neural network for image detection, and the YOLO algorithm [37], employs a fully convolutional neural network, transforming the object detection task into a regression problem that requires only one forward pass to complete the detection and classification of all targets, offering rapid processing. YOLO predicts both the location and category of targets during object detection, effectively avoiding the accuracy loss associated with multiple image scans and excelling in detecting small objects. In the context of internal thread defect detection, this paper adopts YOLO for implementation. The following Table 1 provides a performance comparison analysis of defect detection algorithms.

**Table 1. Performance comparison analysis of defect detection algorithms.**

| Feature | YOLO | Fast R-CNN | SSD |
| --- | --- | --- | --- |
| Accuracy | High | High | Moderate |
| Efficiency | High computational efficiency | less efficient | Efficient |
| Ease of Use | Simple end-to-end architecture | Complex training process | Moderate |
| Real-time Use | Suitable for real-time applications | Less suitable for real-time use | Suitable for real-time use |
| Speed | strong real-time detection capability | Moderate | High |

Particularly, YOLO's capability to concurrently predict target positions and classes during detection minimizes accuracy loss associated with repetitive image scanning and exhibits strong performance in detecting small targets. The necessity for a bespoke dataset, comprising images with corresponding labels and bounding boxes for defect detection, is underscored. Upon the completion of training and validation, YOLO's deployment facilitates real-time object detection and processing, capitalizing on its rapid and precise detection capabilities across various applications. Thus, this project advocates for employing the YOLO algorithm for efficacious internal thread defect detection. The training and deployment process is illustrated in the Fig 7.

## 2.5. Summary

This paper proposes a groundbreaking solution for internal thread defect detection, integrating a cutting-edge hardware setup with advanced image processing techniques. By employing a multi-vision system, augmented with fisheye lenses and strategic LED illumination, we have markedly enhanced image capture fidelity and operational throughput. The deployment of sophisticated algorithms for image stitching, correction of distortion, and defect identification via the YOLO framework has substantially bolstered our ability to detect and scrutinize internal thread anomalies. Our research culminates in a significant leap forward for industrial inspection methodologies, offering a robust, efficient alternative for safeguarding threaded connection integrity.

## 3. Experiment

### 3.1. Vision system based on internal thread image acquisition

In this study, the vision system has been meticulously designed in line with the model analyzed, as depicted in Fig 8. To ensure optimal illumination and mitigate reflection issues associated with internal threading, we employed six fisheye cameras, each integrated with a light source band arranged in a circular configuration. Further enhancing the system's lighting efficacy, a layer of diffusing material was strategically enveloped around the visual apparatus, as illustrated in Fig 9.

### 3.2. Distortion correction effect

Within this study, camera-captured images were employed to identify checkerboard corner points, thereby obtaining corner and diagonal coordinates for sub-pixel optimization. This process involved estimating and calculating the camera's internal parameters, utilizing the sub-pixel optimization principle and adopting the Zhang Zhengyou calibration method to refine calibration outcomes. The optimization of the mapping matrix was continually adjusted based on the camera's internal parameters and distortion vector to establish distortion-free and corrected image relationships. Upon calibration completion, both the internal parameter matrix and distortion parameter matrix were applied to execute precise distortion correction of the images.

This paper utilizes DN100 full-threaded internal screw tubes with an inner diameter of 110 millimeters as the target for collection and detection, illustrated in Fig 10.

Fig 11 showcases the internal thread image post-distortion correction, demonstrating the effectiveness of the applied correction methodologies.

### 3.3. Image preprocessing

This study utilizes an array of filters—mean, median, non-local mean, and 3D block matching—to preprocess images with 10db noise. This selective filtering approach is critical for mitigating noise interference, thereby enhancing the thread images' quality and clarity.

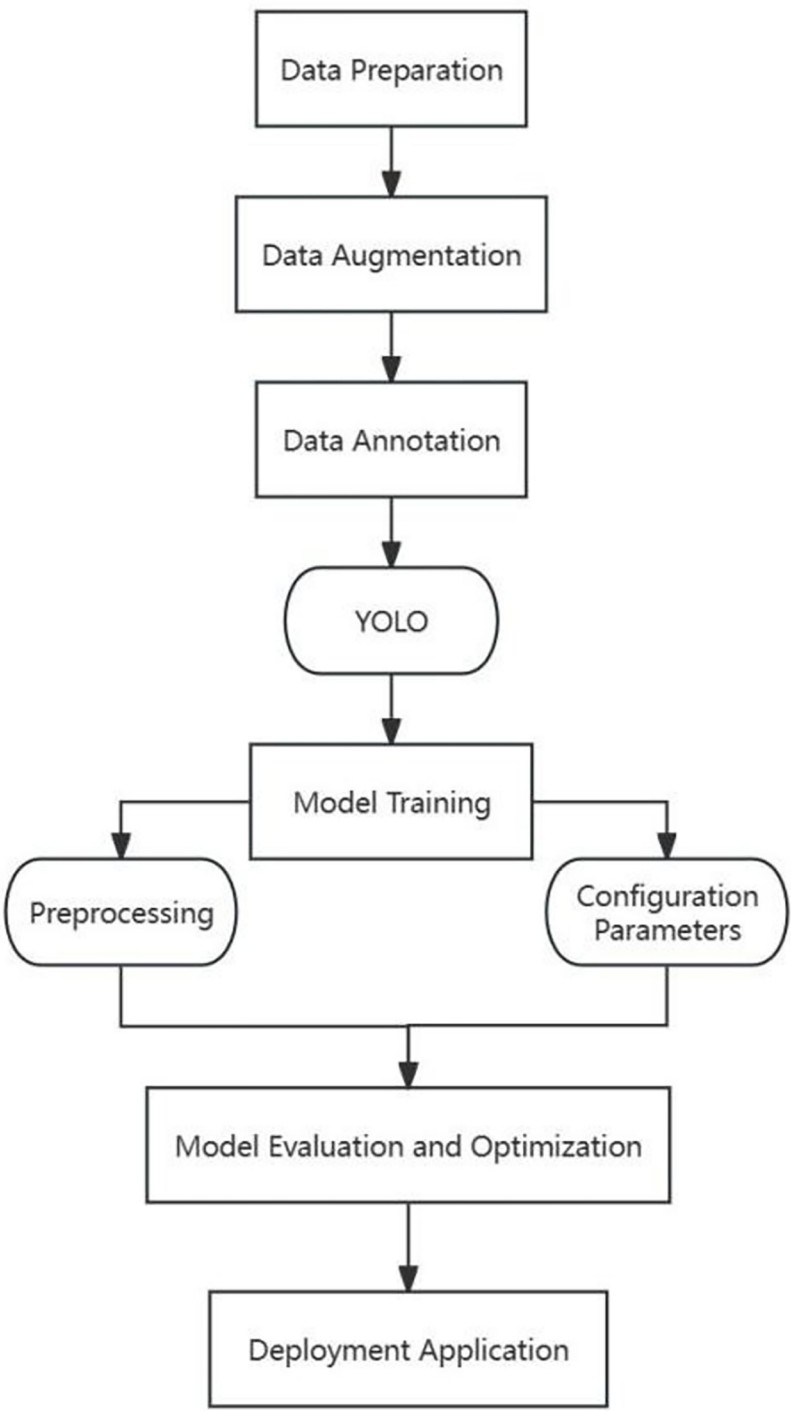

**Fig 7. Training and deployment process.**

Further, image quality enhancement leverages the Retinex algorithm [38], which fine-tunes contrast, color balance, among other image attributes. Additionally, addressing challenges associated with low-light conditions, a specialized low-light image enhancement strategy [39] models and estimates the lighting conditions within images. Subsequent enhancements are

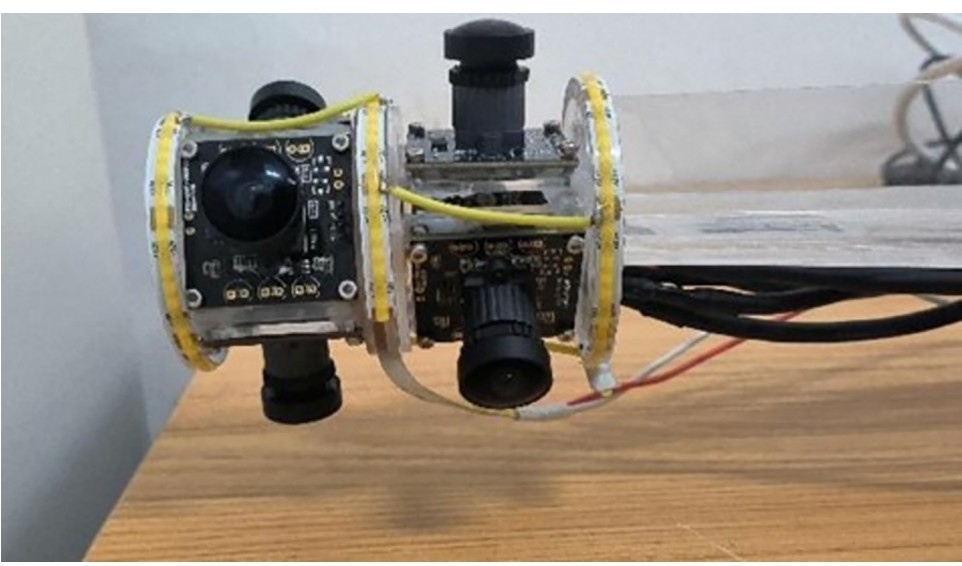

**Fig 8. Visual system photo.**

tailored based on these estimations, significantly boosting image brightness and clarity. Fig 12 exemplifies the outcomes of these image enhancement efforts.

## 3.4. Image stitching

In this study, our database analysis revealed substantial curvature at the margins of the collected images. Attempts to employ splicing algorithms such as Harris, FAST, STFT, SURF, and ORB were made during the image stitching process, yet the outcomes fell short of expectations.

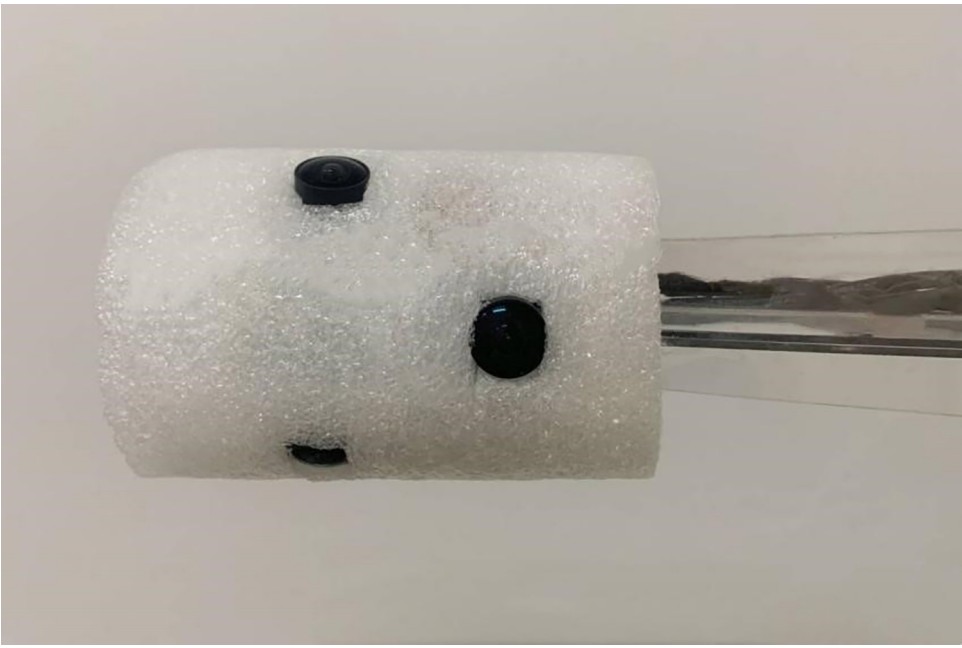

**Fig 9. Visual system improvements.**

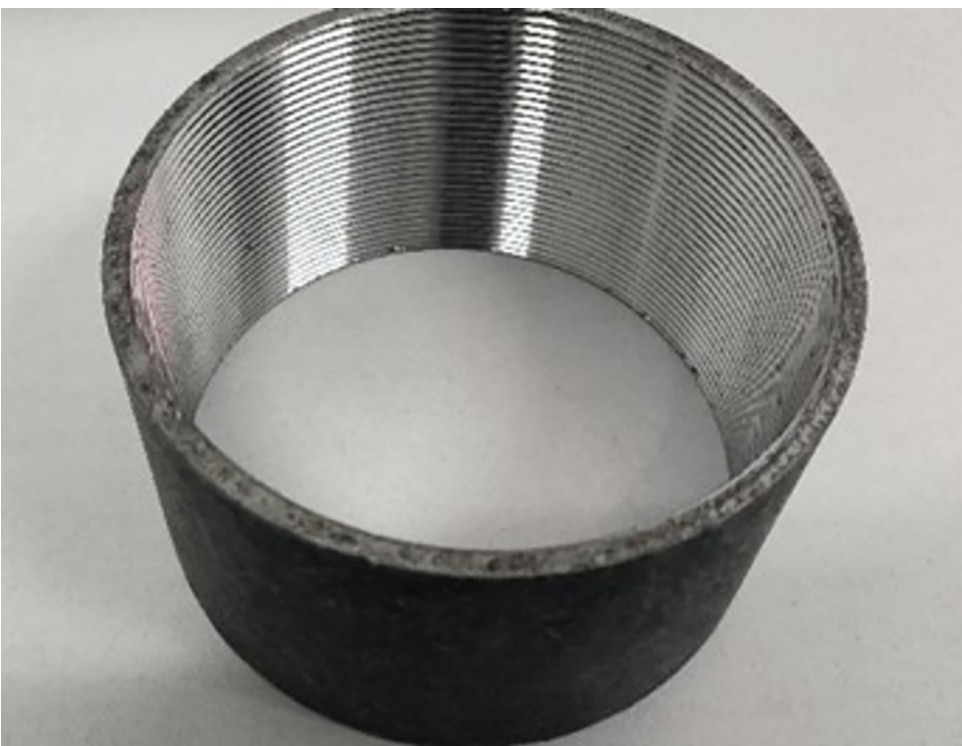

**Fig 10. Tube with female threads.**

Consequently, given the wide viewing angle of our equipment, we opted to strategically crop the acquired images. This entailed removing sections exhibiting pronounced curvature while preserving central portions for effective stitching.

The operational procedure outlined in this paper is as follows: Initial steps involve preprocessing the internal thread images. Subsequently, edge detection algorithms identify the thread line contours. We compute the curvature for each point along the edges within the image, identifying excessively curved regions based on these curvature values. A curvature threshold is established, specific to the internal thread images, and we iteratively assess each edge point's curvature against this benchmark. Points exceeding the threshold prompt the cropping of

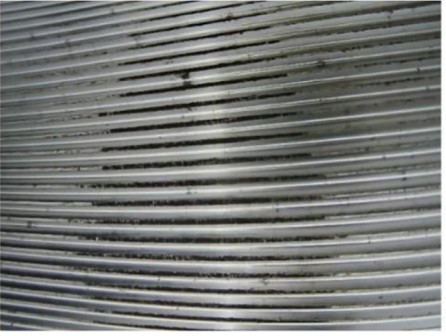
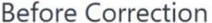

**Fig 11. Internal thread image distortion correction results.**

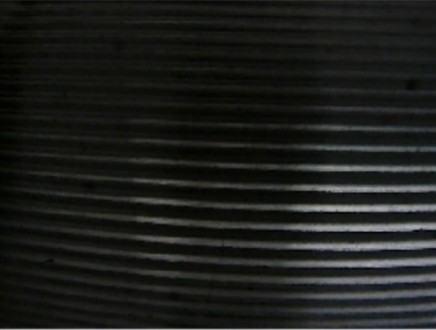 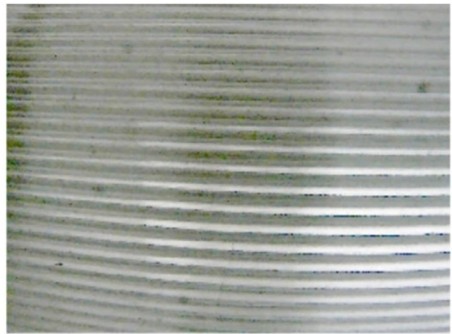

Before Enhancement

After Enhancement

**Fig 12. Internal thread image enhancement results.**

their respective areas. Fig 13 illustrates the images before and after the application of curve clipping.

Post-cropping, the internal threads undergo splicing. To synchronize the repetitive structures and textures inherent to the internal threads, a feature matching method is employed for initial alignment, followed by the application of an image registration algorithm for precise stitch positioning. This process includes the elimination of incorrect matches through filtering and optimization of the match results.

Employing the cylindrical model [40] method, we generate a panoramic image by calculating displacements, converting to a cylindrical coordinate system, and merging the overlapping sections to accurately reconstruct the internal thread's appearance. This study leverages cylindrical mapping technology to ensure seamless image stitching, utilizing the SIFT algorithm for feature detection and the RANSAC algorithm for feature point optimization. Furthermore, an image fusion technique based on Laplace's Pyramid is applied. This involves constructing a Laplace Pyramid for each image to be fused, thereby creating a series of scale-varied sub-images. At each scale, corresponding sub-images are weighted and averaged to produce a fused sub-image. These composite sub-images are then layered to form the final fused image. Notably, the weighted average's parameters can be adjusted to fine-tune the fusion outcome. Fig 14 displays the results of panoramic splicing using this cylindrical stitching approach.

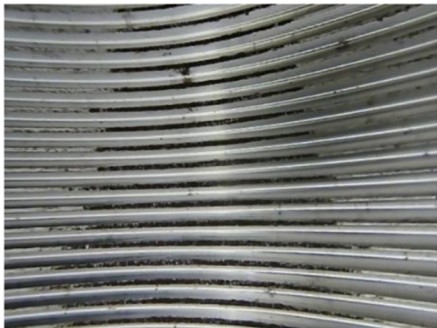 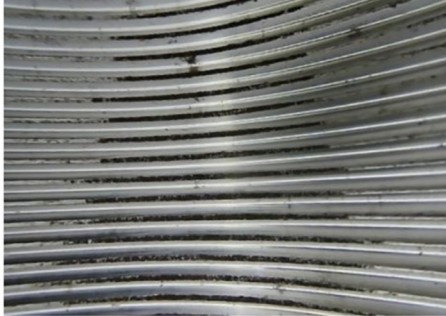

Before Cropping

After Cropping

**Fig 13. Internal thread image cropping results.**

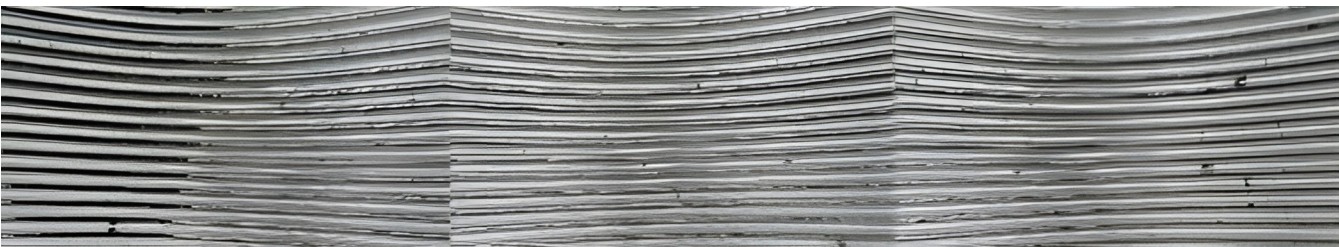

**Fig 14. Panoramic stitching results of internal thread images.**

## 3.5. Defect detection

In the realm of internal thread defect detection, algorithms such as YOLO, SSD, and Fast R-CNN are pivotal, each offering distinct strengths and limitations. This project conducts a comprehensive analysis of these three algorithms' performance, informed by a thorough review [41], to enhance defect detection accuracy.

SSD distinguishes itself by enabling target detection via a single forward pass, thus ensuring rapid processing. Despite its speed, SSD's efficacy diminishes with small object detection, particularly against complex backgrounds or in occluded scenarios. Fast R-CNN, leveraging ROI Pooling for feature extraction, notably boosts both speed and detection precision. Nevertheless, despite advancements over its predecessor, Fast R-CNN does not match YOLO and SSD's speed in large-scale, real-time detection tasks.

When selecting an algorithm for internal thread defect detection, considerations must encompass detection precision, speed, and the capability to identify small targets. YOLOv8 stands out for its exceptional real-time performance and accuracy, making it an ideal choice for scenarios demanding rapid and precise detection. SSD strikes a balance between speed and accuracy, particularly when target object sizes remain relatively constant. Fast R-CNN is preferred in high-accuracy requirements, especially beneficial for smaller datasets or constrained training resources.

Adopting the cutting-edge YOLOv8 algorithm, this project strikes an optimal balance between detection speed and accuracy. YOLOv8's integration significantly elevates both the speed and precision of defect identification, thereby enhancing the detection process's overall efficiency and accuracy. The internal thread detection system, as developed and detailed in this paper, demonstrates substantial practical advantages, boosting detection accuracy and efficiency, markedly lowering false detection rates, and providing robust support for the quality control and safety inspection of internal threads.

## 4. Discussion

The experimental work presented in this study has been successfully concluded, achieving efficient and precise capture of internal thread images. Through meticulous preprocessing, which included image filtering and enhancement, we have effectively preserved essential image information. This foundational work facilitated the seamless stitching of internal thread images, culminating in the successful detection of defects. This process not only underscores the robustness of our methodology but also highlights the potential for significant advancements in the field of internal thread inspection and defect detection.

### 4.1. Image preprocessing

This research meticulously evaluates the efficacy of various filtering technologies, including non-local mean filtering, median filtering, mean filtering, and 3D block matching filtering,

**Table 2. Comparison table of filter algorithm parameters.**

| Feature | Non-Local Means Filtering | Median Filtering | Mean Filtering | Three-Dimensional Block-Matching Filtering |
|---------|--------------------------|------------------|----------------|---------------------------------------------|
| PSNR | 28.5772dB | 24.387dB | 23.5835dB | 34.9441dB |
| SSIM | 0.86163 | 0.82193 | 0.79589 | 0.97136 |

particularly focusing on their impact on image quality. Among these, 3D block matching filtering emerges as superior, notably enhancing the image's Signal-to-Noise Ratio (SNR), as evidenced by improved Peak Signal-to-Noise Ratio (PSNR) values. Furthermore, this method demonstrates exceptional capability in preserving the coherence of the image structure, an advantage quantitatively supported by the Structural Similarity Index (SSIM) values. The comparative analysis of the PSNR and SSIM values processed through these four filtering methods is systematically presented in Table 2. Based on these findings, the 3D block matching filtering technique is identified as the optimal choice for image preprocessing, owing to its dual benefits of elevating image clarity while maintaining structural integrity.

'To further enhance the detail representation and color fidelity of images, this study incorporates an image enhancement technique based on the Retinex algorithm. When compared to Multi-Scale Retinex (MSR) and Single-Scale Retinex (SSR), the Multi-Scale Retinex with Color Restoration (MSRCR) algorithm exhibits superior color restoration and detail definition. This enhancement means that images processed with the MSRCR algorithm appear more natural in terms of color, while also retaining a greater level of detail. Such improvements are instrumental for subsequent phases of image analysis and defect detection.

Post-preprocessing, the spliced images facilitate a two-dimensional expansion of the internal threads. The adjustments in color, brightness, and contrast achieved through filtering and enhancement ensure visual consistency across the stitched image. During the image stitching process, careful attention to the alignment and transition of image edges is paramount to avoid noticeable breaks or discontinuities. This meticulous approach ensures that the final composite image not only presents details and defects with clarity but also maintains a cohesive visual flow.

## 4.2. Image stitching

This study employs cylindrical stitching technology, optimized for the complex geometric structure of internal threads. The cylindrical model-based stitching method is highly suitable for processing images of internal threads, as it adeptly adapts to the annular structure of the threads. When the camera rotates around the axis of the thread during shooting, the cylindrical model effectively reduces image distortion caused by changes in viewpoint, maintaining the geometric characteristics and integrity of the threads. It achieves relatively coherent image stitching in the horizontal direction, particularly suited for generating 360˚ panoramic images. Moreover, cylindrical stitching is less demanding regarding the camera's internal and external parameters compared to the planar model, offering better adaptability and fault tolerance.

In contrast, planar stitching performs excellently in processing wide, flat scenes but tends to produce projection distortions at the edges when dealing with helical or curved internal thread structures, especially in wider stitching areas. The planar stitching method is effective on small-scale, flat, or slightly curved surfaces but is not suitable for objects like internal threads, which exhibit significant geometric variations.

If the projection matrix is calculated through finding corresponding points, using algorithms such as DLT and RANSAC, and then merging multiple photographs into a panoramic image, the unique geometric shape and enclosed space of internal threads may limit the

diversity of viewpoints. Internal thread images have complex textures and repetitive patterns, making it more challenging to precisely match feature points to calculate the homography matrix, thereby leading to significant errors.

The planar stitching method based on ORB, when applied to stitching internal threads, also fails due to the complex geometric structure of the internal threads and distortion errors from fisheye cameras. The features that could have been detected are overlooked due to the abundance of repetitive thread patterns, leading to stitching failures.

The image stitching method used in this paper provides higher accuracy, stability, and visual coherence when processing internal threads, compared to direct planar stitching or simple methods based on feature invariance. Especially for those applications requiring precise maintenance of object geometric characteristics and smooth transition effects, this method significantly outperforms other techniques.

A comparison of the specific stitching effects is illustrated in Fig 15, where A and B represent planar stitching methods, and C represents the stitching method based on the cylindrical model.

## 4.3. Defect detection

This study primarily focuses on conducting an experiment for detecting defects in internal threads using the YOLO algorithm. During the data preparation phase, we meticulously

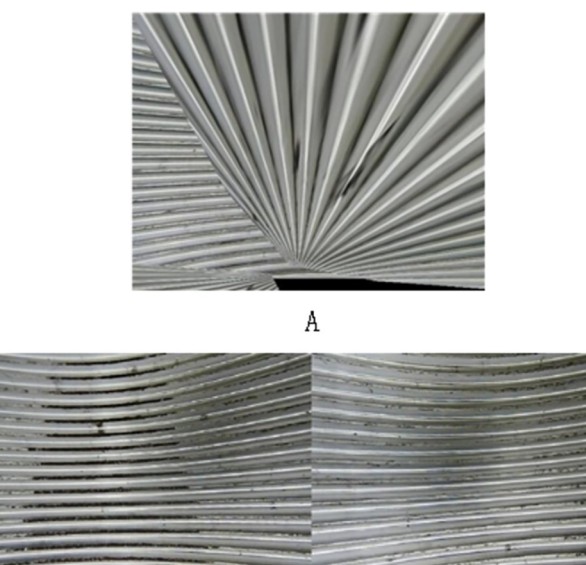

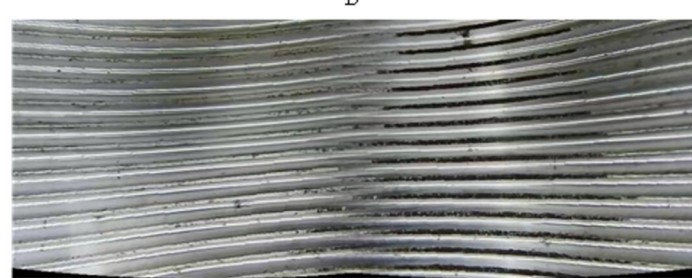

**Fig 15. Comparison chart of image stitching effects.**

collected a diverse array of internal thread images. These images were accurately annotated for defects using professional labeling tools, and the model's generalization capabilities were augmented through data augmentation techniques. YOLOv8, renowned for its high speed and precision, was chosen as the foundational model, with specific optimizations for real-time detection tasks.

The model training procedure encompassed image preprocessing, the setting of training parameters, and the actual training phase. In the subsequent model evaluation and optimization phase, a comprehensive set of metrics, including precision, recall rates, and mean Average Precision (mAP), was employed on the validation set to ensure the model achieved high accuracy and a low rate of false positives. The final, trained model was then integrated into the actual inspection system, enabling real-time, automated detection of defects in internal threads.

This study assesses the efficacy of the proposed model through a self-compiled dataset. The experimental setup was implemented using the Python programming language within the TensorFlow framework, leveraging the computational power of an Nvidia RTX 3090 graphics card. The findings from the experiment indicate that the YOLOv8 model, as introduced in this paper, demonstrates commendable performance in detecting defects within internal threads, achieving the anticipated efficacy.

The Table 3 below compares the training parameters of YOLO v8, YOLOv5, and SSD, quantifying the training outcomes of these three algorithms based on mAP, recall, and precision. Due to limitations in detecting small targets or specific types of defects, SSD has a lower recall rate. YOLOv5 and YOLOv8 show stronger performance in detecting small targets, with YOLOv8 outperforming in terms of accuracy and recall rate, exhibiting superior parameters compared to both SSD and YOLOv5.

In the detection of collected thread images, as shown in Fig 16, defects on the internal threads are marked with red boxes. This project has designated these defects as "corrosion," with the latter being the probability of identifying them as defects, achieving precise detection. The figure below randomly inspects 8 internal thread images (A, B, C, D, E, F, G, H), with a total of 19 defects. Among these, defects in 6 images were completely detected, identifying 17 correct defects in total. The sample average detection rate is 89.47%.

The visualized outcomes are presented in Fig 17, demonstrating the model's effectiveness in accurately identifying defects within internal threads.

Detection poses a significant challenge. The aim of this study is to enhance both the accuracy and efficiency of detecting internal threads by employing a high-resolution camera. In the context of internal thread detection, the physical size represented by each pixel of the camera emerges as a crucial determinant of detection accuracy. This research conducts an in-depth analysis of the relationship between the camera's pixel resolution and its physical size, further analyzing and computing the limiting object resolution distance for the internal thread pipe's pixels in this experiment. The formula provided above serves as the basis for optimizing detection capabilities.

$$\text{Limi Object Resolution per Pixel} = \frac{\text{Thread Diameter}}{\text{Camera Horizontal Resolution(Pixel)}} \quad (1)$$

**Table 3. Comparison of training parameters for YOLO v8, YOLOv5, and SSD.**

| Feature | mAP | recall | precision |
|---|---|---|---|
| SSD | 71.10% | 46.67% | 87.50% |
| YOLOv5 | 91.47% | 73.33% | 86.23% |
| YOLOv8 | 91.56% | 84.68% | 88.36% |

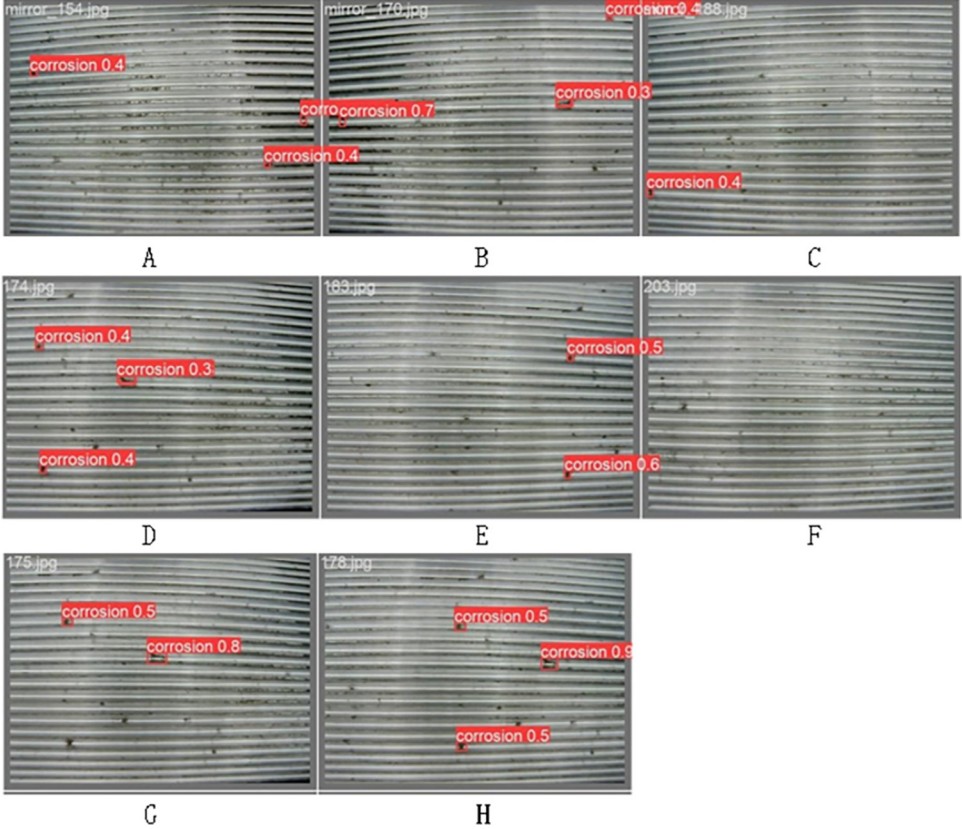

**Fig 16. Internal thread defect detection results.**

In scenarios where detection is conducted on a thread with a diameter of 100 mm utilizing a 1080p resolution camera, the calculated limiting object resolution distance approximates to 0.521 mm. This calculation indicates that, under optimal conditions, the camera is capable of discerning two points separated by a minimum distance of 52.08 microns. For the majority of internal thread inspection tasks, this level of resolution is deemed adequate for conducting inspections with high precision.

## 4.4. Method comparison

The solution proposed in reference [6] is used to measure the measurement area in this paper. This system, utilizing high-precision depth cameras and accurately calibrated sensors, can

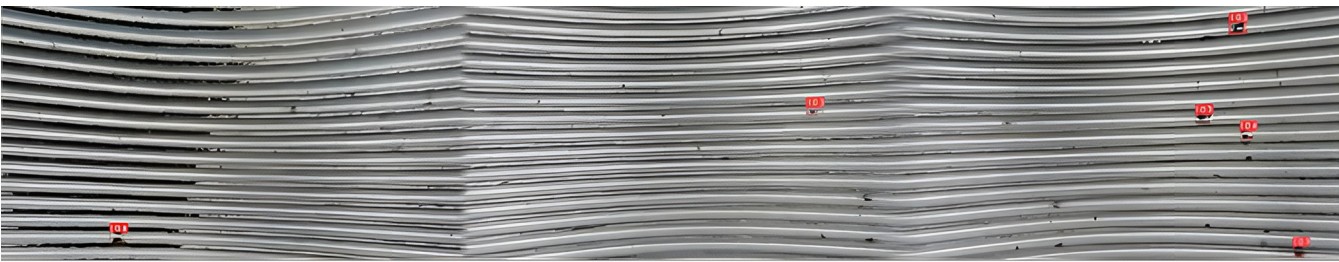

**Fig 17. Defect detection results after internal thread image stitching.**

achieve sub-millimeter measurement accuracy. However, due to the complexity and size of 3D datasets compared to 2D image data, machine vision-based methods typically only involve processing 2D image data, allowing for faster data processing and analysis speeds, enabling real-time detection. The approach presented in reference [8] measures the maximum defect size in the measured area in this paper. While flexible eddy current detection demonstrates high accuracy, machine vision is widely applicable, not limited to metal materials, and can be applied to defect detection in various materials such as plastics, textiles, ceramics, and more.

For the task of detecting corrosion defects in small internal thread targets, a table displays the performance data of three models—SSD, YOLOv5, and YOLOv8—on three evaluation metrics: average precision (mAP), recall, and precision. In the internal thread defect detection task, YOLOv8 exhibits the highest recall and precision, indicating its ability to detect corrosion defects more comprehensively and accurately. Although YOLOv5 performs well in mAP, it falls short compared to YOLOv8 in recall. Meanwhile, SSD performs the worst across all evaluation metrics, especially in recall, possibly due to inherent limitations in the SSD model structure for detecting small targets. Considering the importance of defect detection for safety, the model's recall rate is crucial, making YOLOv8 more suitable for the task of detecting corrosion defects in small internal thread targets. In practical applications, the YOLOv8 model should be prioritized to ensure a high defect detection rate

## 5. Conclusion

This article addresses the challenges related to internal thread detection and proposes an automated detection solution based on machine vision, with the following main contributions:

1. The article designs an efficient image acquisition system for internal thread defect detection by improving lighting conditions and image capture devices, enhancing the quality of internal thread surface images and greatly increasing the efficiency of image acquisition.

2. A method based on a cylinder model for stitching internal thread images is proposed to provide full-field thread images, solving the challenge of stitching such complex textured images.

3. Utilizing the YOLOv8 model, the system accurately locates the positions and sizes of thread defects, demonstrating the effectiveness of this system solution.

## Supporting information

**S1 File.**
(ZIP)

## Author Contributions

**Conceptualization:** Gengpei Zhang.

**Formal analysis:** Xiaohan Dou, Chengqi Xue.

**Funding acquisition:** Chengqi Xue.

**Software:** Zhihao Jiang.

**Supervision:** Gengpei Zhang.

**Writing – original draft:** Xiaohan Dou.

**Writing – review & editing:** Xiaohan Dou.

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
