## [Decision Letter · Decision Letter 0]

5 Mar 2024

PONE-D-24-03693Internal thread defect detection system based on multi-visionPLOS ONE

Dear Dr. Zhang,

Thank you for submitting your manuscript to PLOS ONE. After careful consideration, we feel that it has merit but does not fully meet PLOS ONE’s publication criteria as it currently stands. Therefore, we invite you to submit a revised version of the manuscript that addresses the points raised during the review process.

In particular, one would expect better evidence that the proposed work is competitive when compared to the state of the art. 

We look forward to receiving your revised manuscript.

Kind regards,

Jean-Christophe Nebel, Ph.D

Academic Editor

PLOS ONE

Journal Requirements:

Reviewers' comments:

Reviewer's Responses to Questions

**Comments to the Author**

1. Is the manuscript technically sound, and do the data support the conclusions?

Reviewer #1: Yes

Reviewer #2: Yes

2. Has the statistical analysis been performed appropriately and rigorously? 

Reviewer #1: Yes

Reviewer #2: Yes

3. Have the authors made all data underlying the findings in their manuscript fully available?

Reviewer #1: Yes

Reviewer #2: Yes

4. Is the manuscript presented in an intelligible fashion and written in standard English?

Reviewer #1: Yes

Reviewer #2: Yes

5. Review Comments to the Author

Reviewer #1: This paper proposed a vision-based method that greatly enhances image acquisition efficiency through an advanced hardware architecture. The paper is well organized, and makes a great significance to the current literatures. Few comments are as follows:

1. The contributions of this study should be proposed clearly.

2. The advantages of the proposed approach should be compared to other current approaches, including Image Preprocessing, Image Stitching, especially Defect Detection.

3. Some quantitative results should be proposed in the conclusion section.

Reviewer #2: This manuscript proposes a vision-based approach that significantly enhances the efficiency of image acquisition through sophisticated hardware architecture and addresses particular challenges in industrial inspection, contributing to the field of internal thread defect detection. However, the paper is deficient in innovation, and there is a notable absence of requisite experimental results and analytical discussion. The author should provide additional experimental data to delineate the methodology's advantages and innovations in comparison to alternative techniques for internal thread defect detection.

(1)In the abstract, the authors are encouraged to include empirical data to delineate the superiority of their approach over manual inspection or other existing methods for detecting internal thread defects, as this would lend greater persuasive power to their claims.

(2)Within the introduction, several alternative methods for internal thread defect detection are mentioned, yet there is a lack of detailed discussion regarding their limitations. To better establish the proposed method's advancement, these shortcomings should be elucidated.

(3)The authors may add more state-of-art computer vision articles for the integrity of the manuscript ( Real-Time Defect Detection for Metal Components: A Fusion of Enhanced Canny–Devernay and YOLOv6 Algorithms; Applied Sciences. 3D vision technologies for a self-developed structural external crack damage recognition robot; Automation in Construction.).

(4)The YOLOv8 model, as trained by the authors for the detection of internal thread defects, must have its training outcomes displayed, including key performance metrics such as recall and precision rates.

(5)Figure 17 lacks clarity. It is recommended that the authors re-upload this image with improved resolution or clarity.

(6)The experimental section is wanting in specific data related to internal thread defect identification, such as the count and percentage of detected defects. The inclusion of this data is recommended.

(7)The conclusion of the manuscript should synthesize the experimental findings to provide a cohesive summary of the research's implications and results.

6. PLOS authors have the option to publish the peer review history of their article (what does this mean?). If published, this will include your full peer review and any attached files.

Reviewer #1: **Yes: **Dongming Fan

Reviewer #2: No

---

## [Author Response · Author response to Decision Letter 0]

5 Apr 2024

Dear editor 

We are deeply grateful for the valuable time and thoughtful suggestions provided by the editors and reviewers. Following the guidance of the editors and the feedback from the reviewers, we have made corresponding revisions to the manuscript. These changes are detailed in the "Response to Reviewers" document. We also sincerely apologize for any oversight in the manuscript.

We earnestly hope that this manuscript meets the standards for publication in PLOS ONE. We are immensely grateful for your support throughout this process and eagerly await your response.

Best regards,

Gengpei Zhang

---

## [Decision Letter · Decision Letter 1]

9 May 2024

Internal thread defect detection system based on multi-vision

PONE-D-24-03693R1

Dear Dr. Zhang,

We’re pleased to inform you that your manuscript has been judged scientifically suitable for publication and will be formally accepted for publication once it meets all outstanding technical requirements.

Comments from PLOS Editorial Office: We note that one or more reviewers has recommended that you cite specific previously published works in an earlier round of revision. As always, we recommend that you please review and evaluate the requested works to determine whether they are relevant and should be cited. It is not a requirement to cite these works and you may remove them before the manuscript proceeds to publication. We appreciate your attention to this request.

Kind regards,

Jean-Christophe Nebel, Ph.D

Academic Editor

PLOS ONE

**Comments to the Author**

1. If the authors have adequately addressed your comments raised in a previous round of review and you feel that this manuscript is now acceptable for publication, you may indicate that here to bypass the “Comments to the Author” section, enter your conflict of interest statement in the “Confidential to Editor” section, and submit your "Accept" recommendation.

Reviewer #1: (No Response)

Reviewer #2: (No Response)

2. Is the manuscript technically sound, and do the data support the conclusions?

Reviewer #1: Yes

Reviewer #2: (No Response)

3. Has the statistical analysis been performed appropriately and rigorously? 

Reviewer #1: Yes

Reviewer #2: (No Response)

4. Have the authors made all data underlying the findings in their manuscript fully available?

Reviewer #1: No

Reviewer #2: (No Response)

5. Is the manuscript presented in an intelligible fashion and written in standard English?

Reviewer #1: Yes

Reviewer #2: (No Response)

6. Review Comments to the Author

Reviewer #1: (No Response)

Reviewer #2: (No Response)

7. PLOS authors have the option to publish the peer review history of their article (what does this mean?). If published, this will include your full peer review and any attached files.

Reviewer #1: No

Reviewer #2: No

---

## [Editor Report · Acceptance letter]

14 May 2024

PONE-D-24-03693R1 

PLOS ONE

Dear Dr. Zhang, 

I'm pleased to inform you that your manuscript has been deemed suitable for publication in PLOS ONE. Congratulations! Your manuscript is now being handed over to our production team.

Kind regards, 

on behalf of

Prof. Jean-Christophe Nebel 

Academic Editor

PLOS ONE